# Uncovering Syllable Constituents in the Self-Attention-Based Speech Representations of Whisper

**Erfan A. Shams, Iona Gessinger, Julie Carson-Berndsen**

ADAPT Research Centre, School of Computer Science, University College Dublin, Ireland
{erfan.shams|iona.gessinger|julie.berndsen}@ucd.ie

## Abstract

As intuitive units of speech, syllables have been widely studied in linguistics. A syllable can be defined as a three-constituent unit with a vocalic centre surrounded by two (in some languages optional) consonant clusters. Syllables are also used to design automatic speech recognition (ASR) models. The significance of knowledge-driven syllable-based tokenisation in ASR over data-driven byte-pair encoding has often been debated. However, the emergence of transformer-based ASR models employing self-attention (SA) overshadowed this debate. These models learn the nuances of speech from large corpora without prior knowledge of the domain; yet, they are not interpretable by design. Consequently, it is not clear if the recent performance improvements are related to the extraction of human-interpretable knowledge. We probe such models for syllable constituents and use an SA head pruning method to assess the relevance of the SA weights. We also investigate the role of vowel identification in syllable constituent probing. Our findings show that the general features of syllable constituents are extracted in the earlier layers of the model and the syllable-related features mostly depend on the temporal knowledge incorporated in specific SA heads rather than on vowel identification.

## 1 Introduction

Syllables have long played a central role in phonological theory and are relatively more intuitive to grasp than other phonological entities such as segments (Hayes, 2009). Moreover, syllables represent a language-specific systematic organisation of sounds which allow native speakers of a language to differentiate between well-formed sequences of sounds which may not constitute actual words of the language, and ill-formed sequences which are not permissible in that language. For example, in English, the word *blick* is considered well-formed (an accidental gap in the lexicon) whereas

*bnick* is ill-formed (a systematic gap); this is because the syllable constituent *bl* exists at the beginning of a syllable in English but *bn* does not. This has inspired many automatic speech recognition (ASR) developments based on syllables, both in the past (Bartels and Bilmes, 2007) and the present (Anoop and Ramakrishnan, 2023). One of the main arguments in earlier ASR model design was that the use of syllables would offer a limited number of sub-words which in turn makes the coding of the model more efficient (Scharenborg et al., 2005). Even though n-gram-based byte pair encoding (BPE) offers a more simplistic approach to a language-agnostic solution, humans can understand syllables as a unit of speech much better than n-grams. Furthermore, the results from Anoop and Ramakrishnan (2023) show that syllable-based BPE and unigram-language modelling can offer better performance when coupled with a conformer (Gulati et al., 2020) speech encoder and transformer (Vaswani et al., 2017) language decoder.

Transformer and conformer architectures rely heavily on the self-attention (SA) weights optimised during the training. The learned parameters in the SA heads define the characteristics of each SA head. One of the core functionalities of the SA mechanism is that it takes the positional dependencies of the input to the output into account, e.g., mapping a segment of the input audio signal to the phonetic localisation of the embedded frame (Shim et al., 2022). Given a sufficiently large amount of training data, transformer-based models achieve a high performance. However, the interpretability of the model is not given by design. This leaves the question of whether these models organise sounds systematically into well-formed syllables similar to native speakers of a language, or whether they contextualise based on the acoustic features of the audio alone. Previous results suggest that the SA weights can contribute

to syllable-based ASR. For example, Moriya et al. (2020) demonstrated that distilling SA weights for building connectionist temporal classification-based ASR reduces character/kana-syllable error rates for Japanese. Other recent work by Zhou et al. (2018) shows that a transformer module incorporated into a syllable-based ASR is superior to context-independent phoneme-based models for Mandarin. This implies that SA weights attend to acoustic/contextual information needed for identifying syllables.

Although methods such as SA weight distillation (Moriya et al., 2020) hint at the ability of large black box models to extract relevant features for syllable-based ASR systems, they do not demonstrate *where* and *how* the relevant features are embedded. In this paper, we explore an approach which evaluates the SA weights and consequently the latent representations (also known as embeddings) of OpenAI's transformer-based Whisper (Radford et al., 2023) via SA head pruning combined with domain-informed probing tasks. First, we measure the capacity of the model to capture the distinctive features of the syllable constituents *onset*, *nucleus*, and *coda* in the English language (see Section 2.1) and to the phonetic categories *vowel*, *consonant*, and *silence*. Second, we explore the relevance of the phonetic categories in identifying the syllable constituents by using an SA head pruning method. Figure 1 illustrates the overall workflow of this study which is explained in Section 3. We find that the SA heads in the initial layer of the transformer model extract the general features needed for both probes (syllable constituent and phonetic category probe) and thus pruning any of the SA weights from this layer has a much higher impact than pruning weights from the SA heads in the other layers. However, not all SA heads contribute equally to encoding these features. Even where we expect an overlap (e.g., nucleus and vowel), we find that syllable constituents and phonetic categories are not necessarily contextualised in the same way.

The paper first provides background on syllable constituents and ASR probing in Section 2. It then gives details about the probed models, the speech corpus, and the probing tasks, as well as the SA pruning method in Section 3. Section 4 assesses the probe results and the impact of the SA pruning on syllable constituent encoding. The conclusion and future work are detailed in Section 5, while limitations of this work are discussed in Section 6.

## 2 Background

Linguistic studies surrounding syllables and ASR probing are the two key motivators for the work presented in this paper. We provide further information on these concepts in the following sections.

### 2.1 Syllable Constituents

While there is much debate about the exact definition of a syllable, and how to determine the number of syllables or the location of syllable boundaries, they do constitute a fundamental unit of speech perception (Mehler et al., 1981) and production (Browman and Goldstein, 1988), and are often intuitively accessible to humans (Ladefoged and Johnson, 2010). A syllable ($\sigma$) can be described as consisting of the following three constituents:

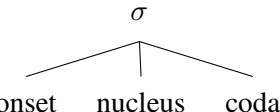

with the *nucleus* as the vocalic centre of the unit, and the *onset* and *coda* comprising all consonants before and after the vowel respectively (Ladefoged and Johnson, 2010). In other words, every vowel forms the centre of a syllable, while onset and coda are optional. To determine syllable boundaries, the principle of onset maximisation is often applied, which suggests that consonants are preferably assigned to the onset of the following syllable rather than the coda of the preceding one (Selkirk, 1982). Whether the allocation of a consonant to the onset is permissible depends on the phonotactic rules of a given language, i.e., which sounds can follow one another in order to be well-formed in that language (Hayes, 2009). Therefore, syllable structure and complexity of onset and coda vary considerably between languages, which makes the automatic segmentation of syllable constituents a non-trivial problem.

### 2.2 ASR Probing

Deep learning models are infamous for being opaque when interpreting their decision-making process (Becker et al., 2018). Post-hoc explainable-AI (XAI) methods including domain-informed probing tasks are a viable approach to this issue. Probing transformer-based models, especially in NLP is an ongoing endeavour (Conneau et al., 2018; Nedumpozhimana and Kelleher, 2021; Klubička et al., 2023).

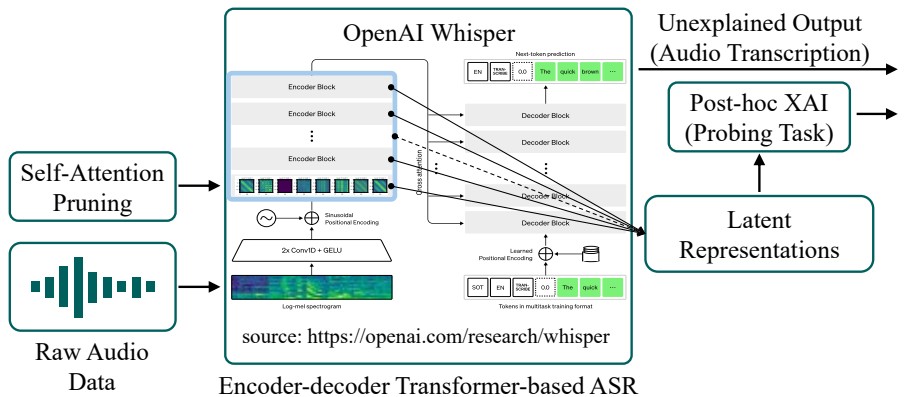

Figure 1: Probing task and self-attention head pruning on OpenAI's transformer-based Whisper model.

Probing ASR models in particular started gaining popularity with works like Shah et al. (2021) inspiring the probing methodology in the present study. Probing tasks in ASR have been explored in recent years for tracing the capacity of the modern ASR models to encode phonetic (English et al., 2023, 2024) and prosodic (Yang et al., 2023) information. English et al. (2023) found that the embeddings of the transformer-based wav2vec 2.0 (Baevski et al., 2020) encode distinctive information related to the physical constraints of feature co-occurrence mostly in higher layers of the model. In other words, the independently trained probes for nasal, fricative and voicing features tend to erroneously detect nasal and fricative co-occurrences more frequently in the lower layers compared to the higher layers such as layer 9. The majority of the erroneous cases of co-occurrence detection were circumstances with nasal and fricative sounds in close proximity. We further analysed how the transformer-based embeddings capture the presence of different articulatory phonetic features based on international phonetic alphabet (IPA) classification (English et al., 2024). Through domain-informed probing, we found that the articulatory features are captured best in higher layers of the model. Additionally, probing for articulatory features allowed us to see subtle changes in place and manner of articulation where for instance, the probes were able to detect epenthesis during transition from a bilabial-nasal phone to a labiodental-fricative phone. We further investigated articulatory feature overlap in consonant clusters where a complete or partial feature overlap is expected (Shams et al., 2024). The joint probabilities of independently probed place and manner of articulation and voicing suggest the presence of alternative articulatory features influenced by the surrounding

sounds. The probabilities of the probe outputs are also used by de Heer Kloots and Zuidema (2024) to investigate the phonotactic constraints of the English language embedded in the latent representations of wav2vec 2.0. They showed that the phonotactic bias towards existing consonant clusters in English is present in the higher layers.

While our past studies focused on the segmental level, Bentum et al. (2024) showed that the abstract contextualisation in the higher layers of the wav2vec 2.0 steers away from solely relying on the segment-level acoustic features in the identification of stress which is considered a suprasegmental feature. That is to say, while the early convolution layers of the model represent stress in a segmental manner, the higher transformer layers extract a more generalised representation of stress based on the surrounding context of a vowel. This was shown by training the stress classifiers while leaving out a specific vowel or including only a particular vowel. This way, the difference in stress classification performance between the two sets shows that the more generalised representations of the higher transformer layers are less affected (perform the same on both sets) compared to the codevectors which are mapped from the output of the convolution layer (perform worse when given only the left-in vowel set).

Furthermore, a recent paper by Vitale et al. (2024), specifically explored the probing task of identifying syllable boundaries across the latent representations of Nvidia's English-only conformer-based NeMo model. Three versions of the model with different parameter sizes were probed. The authors used Spanish and Italian speech corpora to extract the latent representations. They concluded that the lower layers of the model encode the rhythmic information needed for identi-

fying the syllable boundaries which is similar to our finding for syllable constituents. They also mentioned the potential of the extracted representations for training smaller ASR models with competitive results.

While the above mentioned works focus on the latent representations of the models, a study by Mohebbi et al. (2023) leverages context mixing methods to evaluate the attention weights of specific tokens in connection with homophony in French. SA weights can also be evaluated using the averaging method where an SA head is replaced by a static version obtained from averaging its activation values on a corpus (Hassid et al., 2022). A static SA can also be an identity matrix where the attention from one frame is only *to* and *from* itself. In addition, the attention can be set to zero. This method is known as pruning. Pruning can be used to detect redundant layers in ASR models which can potentially speed up the inference time by ablation while retaining the overall performance of the model (Wang et al., 2023). The SA head pruning technique was also explored in our recent study of articulatory feature probing in Whisper (Shams and Carson-Berndsen, 2024). We demonstrated a use case of this approach in identifying SA heads which contribute to encoding certain features for distinguishing between alveolar and postalveolar sounds within an utterance. In this case, pruning a certain SA head, identified through visual inspection, increased confusion between the two sounds.

The study presented in this paper sheds further light on *where* and *how* syllable-related information is embedded in the latent representations of Whisper by jointly probing them for syllable constituents and phonetic categories, by focusing on the role of SA heads. Since the vocalic portion of a syllable always constitutes the syllable nucleus, we anticipate that the syllabic constituent nucleus would be encoded in the same way as the phonetic category vowel. In the next section, we set out the materials and methods used in this study.

## 3 Materials and Methods

The details of the overall workflow depicted in Figure 1 are presented in the following sections.

### 3.1 Models and Corpus

The models evaluated in this paper are OpenAI's transformer-based encoder-decoder ASR models *whisper-base* (multilingual) and *whisper-base.en*

(English-only).[1] Both models have 74 million parameters in total with 6 layers, and 8 SA heads in each layer for both encoder and decoder blocks. However, their training data and trained parameter values including the SA head weights differ. These models were chosen to assess whether there are any effects on English syllable identification due to the variety of languages in the training set.

To extract and label the latent representations of the ASR models we require an English speech corpus with time-aligned annotation regardless of the performance of the models in transcribing the utterances. A domain-informed probing task aims to evaluate the layerwise capacity of a model in encoding domain-specific information rather than the word error rate (WER) performance. Hence, choosing a corpus with expert-annotated time-aligned phone and word-level labels is important for this particular study. The latent representations of each encoder layer are extracted using utterances from the TIMIT corpus (Garofolo et al., 1993). TIMIT is an English language corpus with 5.4 hours of read speech by 630 speakers. The time-aligned phonetic and word (orthographic) transcriptions of this corpus are used to extract feature labels for syllable constituents and phonetic categories. For converting the TIMIT timestamps into the Whisper model timestamps, the former are divided by 320 and rounded to the nearest integer, since 320 is the fixed value for the number of audio samples per model frame in all Whisper models. The latent representations in each layer are stored in a $1500 \times 512$ tensor, where 1500 is the number of frames corresponding to the padded 30 second input audio, i.e. the required audio input length. We discard the padded frames by calculating the valid frame length based on the total number of samples in the input audio and the 320 audio samples per frame mentioned above. Mean aggregation is then used to reduce multiple consecutive frames corresponding to the same phone into a single-frame representation. For instance, we average the latent representations of $n$ number of frames annotated as a certain vowel.

Syllables are identified using an English syllabifier.[2] By default, this syllabifier uses the standard phonetic transcription of the CMU pronunciation dictionary.[3] However, here we employ the concrete

---

[1] https://github.com/openai/whisper
[2] https://github.com/emmaon/syllabifier/blob/master/code.py
[3] http://www.speech.cs.cmu.edu/cgi-bin/cmudict

phonetic transcription of TIMIT, mapping the original 60-phone system into the 39-phone system[4] plus the glottal stop. Within the identified syllables, the vowel is labelled as nucleus and frames which come before or after a vowel in a syllable are labelled as onset or coda respectively (see structure described in Section 2.1). For the phonetic categories, the frames are labelled as vowel, consonant, or silence based on their corresponding TIMIT phone annotation.

## 3.2 Probes

A domain-informed probing task involves training a relatively simple machine learning model on the latent representations of a more complex model for a domain-specific task. Simple model architectures such as multi-layer perceptrons with only one hidden layer and a limited number of hidden units highly depend on the quality of the input and do not extract deeper features. Therefore, it is a viable approach to identify the relevance and capacity of a model's embeddings with respect to a certain domain.

In this study, we trained 24 probes (12 constituent and 12 category probes corresponding to the encoder layers of the *whisper-base* and *whisper-base.en* models combined) on the labelled representations explained in Section 3.1. The probes are based on a simple multilayer perceptron (MLP) architecture by scikit-learn[5] with 512 inputs corresponding to the number of features in each frame of the representations, one hidden layer with 200 ReLU activated neurons, and 3 outputs corresponding to the above mentioned classes of each probe (onset, nucleus, and coda for the constituent probe; consonant, vowel, and silence for the category probe). The activation function of the probe outputs is softmax, the maximum number of training epochs is set to 200, and all other hyperparameters are left as default.

Probe performance is assessed in terms of individual class recall and overall accuracy throughout the paper. The individual class recall gives a better insight into the capacity of the probes to identify each constituent, while the overall accuracy of the probes measures the impact of SA head pruning.

The class $i$ recall ($Recall_i$) is calculated by Equation 1, while the overall accuracy ($Accuracy$) is calculated by Equation 2.

---

[4] https://github.com/kaldi-asr/kaldi/blob/master/egs/timit/s5/conf/phones.60-48-39.map

[5] https://scikit-learn.org/

$$Recall_i = \frac{TP_i}{TP_i + FN_i} \qquad (1)$$

where $TP_i$ is the number of true positive predictions and $FN_i$ is the number of false negative predictions of class $i$.

$$Accuracy = \frac{TP + TN}{TP + TN + FP + FN} \qquad (2)$$

where TP, TN, FP, and FN are true positive, true negative, false positive and false negative predictions of all classes.

## 3.3 Self-Attention Head Pruning

The attention mechanism in transformer networks is known as scaled-dot product attention which takes the input vector (processed audio signal in ASR) as query, key, and value vectors, and calculates the attention weights by performing a dot product of query and key modified by a factor of the input key dimension, cf. Vaswani et al. (2017). In a multi-head attention architecture, a fixed number of SA heads work in parallel to attend to various aspects of the input. The calculated attentions are then concatenated and projected into a linear vector fed into the feed-forward section of the layer output processing block. Vaswani et al. (2017) explained in their original report that the presence of multiple SA heads working in parallel would allow the model to attend to different representations of the input in distinct positions which is an advantage compared to a single attention head. However, as mentioned in Section 2, not all SA heads might be contributing equally to the inference of the model; some may even be redundant.

To measure the impact of different SA heads on the probing performance and to confirm whether the nucleus identification in the constituent probe relies directly on the encoded vowel information, we use weight zeroing which sets all learned parameters of a certain SA head to zero, in other words prunes it. This will nullify the effect of the pruned SA from the latent representations of its layer. We prune one of the eight SA heads per layer at a time, extract the latent representations for the current and subsequent layers, and then evaluate the probes on the new latent representations.

## 4 Results

The experimental results are presented in the following four sections including the probing result

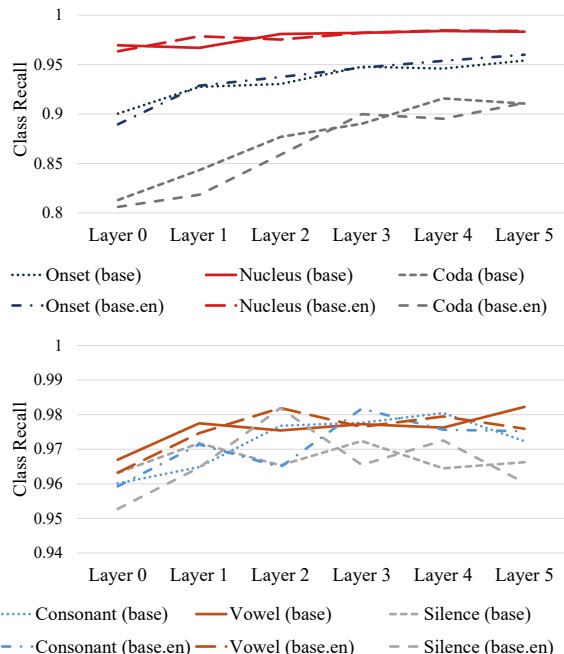

Figure 2: Constituent (top) and category (bottom) probe performance in terms of class recall on TIMIT test data for the *whisper-base* and *whisper-base.en* models.

Table 1: Syllable constituent probe accuracy after SA head pruning compared to the baseline (BL). Attention types identified for layer 0 (see Section 4.4) are highlighted: temporal (blue), phone-based (red), and hybrid (grey) attention.

| | | **Layer** | | | | | |
| | | **0** | **1** | **2** | **3** | **4** | **5** |
| | | *whisper-base* | | | | | |
| **BL** | | 0.911 | 0.927 | 0.940 | 0.950 | 0.956 | 0.957 |
| **Pruned SA head** | 0 | 0.850 | 0.922 | 0.926 | 0.948 | 0.955 | 0.958 |
| | 1 | 0.652 | 0.902 | 0.940 | 0.949 | 0.952 | 0.957 |
| | 2 | 0.874 | 0.892 | 0.920 | 0.949 | 0.946 | 0.954 |
| | 3 | 0.698 | 0.921 | 0.939 | 0.946 | 0.955 | 0.957 |
| | 4 | 0.694 | 0.914 | 0.939 | 0.950 | 0.955 | 0.955 |
| | 5 | 0.830 | 0.924 | 0.940 | 0.944 | 0.952 | 0.958 |
| | 6 | 0.720 | 0.918 | 0.939 | 0.937 | 0.945 | 0.956 |
| | 7 | 0.452 | 0.920 | 0.919 | 0.938 | 0.955 | 0.949 |
| | | *whisper-base.en* | | | | | |
| **BL** | | 0.903 | 0.927 | 0.937 | 0.952 | 0.954 | 0.960 |
| **Pruned SA head** | 0 | 0.659 | 0.908 | 0.917 | 0.952 | 0.954 | 0.960 |
| | 1 | 0.854 | 0.918 | 0.931 | 0.937 | 0.952 | 0.959 |
| | 2 | 0.539 | 0.924 | 0.934 | 0.952 | 0.954 | 0.955 |
| | 3 | 0.821 | 0.920 | 0.935 | 0.949 | 0.952 | 0.959 |
| | 4 | 0.887 | 0.912 | 0.917 | 0.952 | 0.938 | 0.953 |
| | 5 | 0.891 | 0.924 | 0.936 | 0.950 | 0.954 | 0.960 |
| | 6 | 0.719 | 0.910 | 0.936 | 0.938 | 0.951 | 0.961 |
| | 7 | 0.699 | 0.897 | 0.937 | 0.949 | 0.954 | 0.962 |

(Section 4.1) followed by the impact of SA head pruning on syllable constituent (Section 4.2) and vowel/nucleus identification (Section 4.3), and finally the effect of different SA head types on syllable constituent identification (Section 4.4). Additionally, graphs showing detailed results for all SA head pruning constellations in *whisper-base* (Appendix A) and *whisper-base.en* (Appendix B) are available on the OSF wiki page of this project.[6]

### 4.1 Probing

The results of the probing tasks are shown in Figure 2. For the syllable constituents (top), class recall is above 80% in all cases and generally increasing in each layer, especially for onset and coda. This suggests that, while all layers encode the required information for identifying the nucleus (the highest performing feature), the later layers encode information which helps differentiate between all three constituents more efficiently. For the phonetic categories (bottom), all layers have a class recall above 95%, while layers 2 and 3 show the highest overall performance with around 98% class recall for all categories.

---

[6] https://osf.io/s9d2h/wiki/home/?view_only=17f8c2f53f1241958d636af3b656817b&view

### 4.2 SA and Syllable Constituent Accuracy

The self-attention head pruning process is carried out for constituent and category probes on both *whisper-base* and *whisper-base.en*. The outputs of both probing tasks are then analysed separately using the overall accuracy as well as the individual class recall. Table 1 includes the constituent probe performance on the latent representation for each pruned SA head. The accuracy of the baseline (BL) probe (without SA head pruning) is also given in the first row for each model. For instance, when SA head 0 for layer 0 (denoted by $H_{0,0}$ of the *whisper-base* model is pruned, the constituent probe accuracy drops from 0.911 in BL to 0.850, and for SA head 7 of layer 0 ($H_{0,7}$) the accuracy drops to 0.452. Comparing performance after the SA head pruning with the BL performance, we can see that pruning in the earlier layers, especially layer 0, has a higher impact than pruning in the later layers. Among the SA heads in layer 0, pruning $H_{0,7}$ and $H_{0,2}$ for the *whisper-base* and *whisper-base.en* models respectively have the highest impact on the constituent probe performance.

Looking further into the impact of pruning on individual syllable constituents, we observe that different SA heads have different effects on onset,

nucleus and coda identification. For instance, in the *whisper-base.en* model, pruning $H_{0,1}$, which reduces the overall accuracy of the probe for layer 0 by about 5%, has a small impact on the nucleus while true positive predictions for the coda increase at the cost of true positive predictions for the onset. On the other hand, pruning $H_{0,2}$ of the same model reduces the constituent probe performance drastically by about 67% which is mostly due to a true positive drop for both onset and nucleus. To understand whether the constituent probe is affected by the functionality of SA heads in encoding vowel details for nucleus detection, we look into the performance impact on the phonetic category probe for each layer as well.

### 4.3 SA in Vowel and Nucleus Identification

Referring to the results presented on the OSF wiki page of this project (see *Vowels + Syllable Constituents* in Appendices A and B), pruning any SA head in layer 0 affects the recall of the vowel category consistently more than the consonant and silence categories. This indicates that all SA heads contribute to encoding the information required for the probe to distinguish between a vowel and other categories.

Further analysis of the impact of SA heads on both probes in layer 0 of *whisper-base.en* do not suggest a direct relation between the nucleus and vowel recall. For instance, Figure 3 compares the performance between the individual features of both probes for pruned $H_{0,2}$ and $H_{0,1}$ SA heads. The graph indicates that while pruning $H_{0,1}$ has minimal impact on nucleus identification, it has markedly more effect on vowel identification. On the contrary, pruning $H_{0,2}$ shows an almost equal impact on both nucleus and vowel identification while the true positive degradation for vowel is less than when $H_{0,1}$ is pruned.

### 4.4 SA Head Types

Observing no direct connection between the vowel and nucleus identification, we looked into the SA heads for further explanation. Figure 4 illustrates the attention weights for each SA head of layer 0, with higher attention weights appearing in a brighter, yellow colour. We can see two major patterns in attention to the encoded frames. In the first type (blue; see SA heads 1, 3, 4, and 6 of *whisper-base*; 0, 6, and 7 of *whisper-base.en*), the attentions are uniformly distributed on the diagonal, attending to the current frames and the closest neighbours.

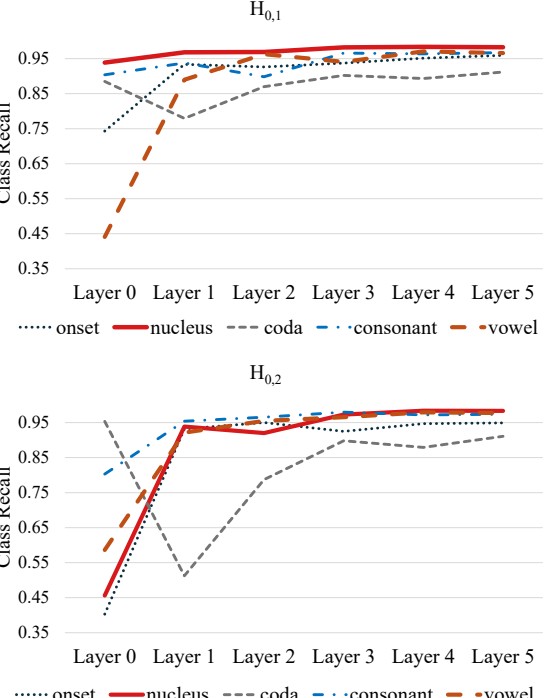

Figure 3: The effect of $H_{0,1}$ and $H_{0,2}$ SA head pruning on constituent and category probe class recall in *whisper-base.en*.

We refer to this as *temporal* attention. In the second type (red; see SA heads 0, 2, and 5 of *whisper-base*; 1, 3, 4, and 5 of *whisper-base.en*), the attentions are selectively activated on different frames. We refer to this as *phone-based* attention. Additionally, SA heads 7 of *whisper-base* and 2 of *whisper-base.en* show properties of phone-based and temporal attention. We refer to this as *hybrid* attention (grey). Table 1 shows that the hybrid-type SA heads (grey) have the most impact on constituent probe accuracy, followed by the temporal attention (blue). The phone-based attention (red) turns out to have the least impact.

To quantify the relationship between temporal attention weights and the accuracy after pruning, we compute the diagonality score (DS) of all SA weights after softmax in layer 0 for the entire test set using formula (3) from Yang et al. (2020), and calculate the Pearson correlation coefficient (PCC) between the obtained DS and the accuracy after pruning. The computed DS presented in Table 2 closely matches the visual inspection of the SA heads in Figure 4, with the hybrid-type SA heads displaying a score between the temporal and phone-based attentions. Furthermore, we calculated the PCC with and without the hybrid attentions (PCC+hybrid and PCC−hybrid, respec-

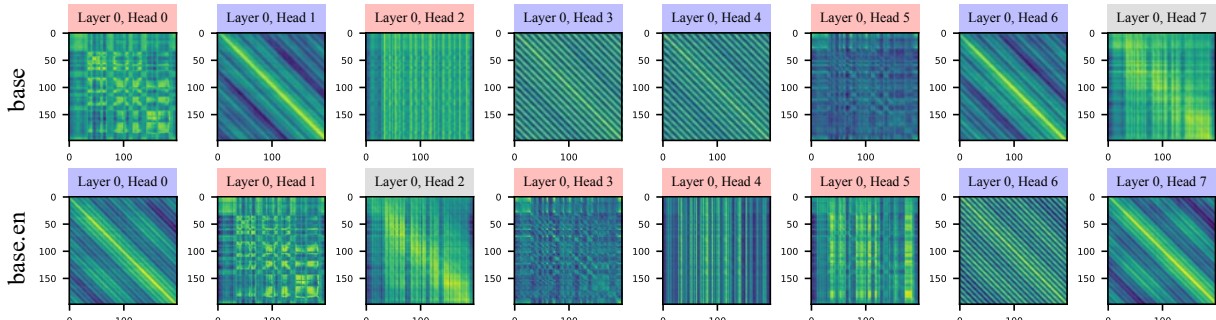

Figure 4: Self-attention heads in *whisper-base* (top) and *whisper-base.en* (bottom) before softmax. Identified attention types are highlighted: temporal (blue), phone-based (red), and hybrid (grey) attention.

Table 2: Pearson correlation coefficient (PCC) between accuracy after pruning and diagonality score (DS) of SA heads in layer 0 of *whisper-base* and *whisper-base.en*. Identified attention types are highlighted: temporal (blue), phone-based (red), and hybrid (grey) attention.

| | | *whisper-base* | | *whisper-base.en* | |
|---|---|---|---|---|---|
| **Layer 0** | | **Accuracy** | **DS** | **Accuracy** | **DS** |
| | **0** | 0.850 | 0.801 | 0.659 | 0.983 |
| | **1** | 0.652 | 0.975 | 0.854 | 0.815 |
| **Pruned SA head** | **2** | 0.874 | 0.758 | 0.539 | 0.862 |
| | **3** | 0.698 | 0.993 | 0.821 | 0.856 |
| | **4** | 0.694 | 0.992 | 0.887 | 0.731 |
| | **5** | 0.830 | 0.828 | 0.891 | 0.757 |
| | **6** | 0.720 | 0.976 | 0.719 | 0.993 |
| | **7** | 0.452 | 0.847 | 0.699 | 0.978 |
| **PCC+hybrid** | | -0.407 | | -0.657 | |
| **PCC−hybrid** | | -0.967 | | -0.970 | |

tively). The results in Table 2 show that while in general, there is a moderate negative correlation between the accuracy after pruning and DS, excluding the hybrid-type attentions (grey) increases the strength of the negative correlation. In other words, when exclusively comparing SA heads with temporal and phone-based patterns, removing temporal SA heads is more detrimental to the identification of the syllable constituents.

## 5 Conclusion and Future Work

In this study, we probed OpenAI's *whisper-base* multilingual and English-only versions for the syllable constituents onset, nucleus, and coda. The probing results show that the earlier layers of the models already encode the information required to identify syllable constituents, while the later layers improve on this by encoding more relevant features. We observed no substantial differences between the English-only (*whisper-base.en*) and the multilingual (*whisper-base*) versions; this could be due to the majority of the data being English speech for

training the multilingual model. Additionally, the models were probed for phonetic categories (vowel, consonant, and silence) to assess whether there is any connection between identifying a vowel and the nucleus of a syllable. To that end, a self-attention head pruning technique known as zeroing was used in conjunction with the probing tasks. This allowed us to identify the impact of different types of self-attention weight patterns on the embeddings.

While pruning the SA heads impacted the performance of the probes to identify a syllable nucleus and vowels, the results showed no direct connection between the two features. However, we found that pruning SA heads with a hybrid temporal and phone-based attention pattern decreased the accuracy of syllable constituent identification more compared to SA heads with purely temporal attention patterns. This was confirmed by calculating the Pearson correlation coefficient between the accuracy after pruning and the diagonality score of SA heads. Overall, our findings imply that the temporal location of the self-attention weights is a more impactful factor in probing syllable constituents than purely phone-based weights.

This approach can be particularly valuable in identifying relevant SA heads which can be used for SA distillation in designing syllable-based ASR models, similar to what Vitale et al. (2024) suggested regarding distillation of latent representations. In our case, this would involve utilising the relevant SA weights from a larger model as a teacher for a new model.

In our future work, we will further study syllable constituents in the scope of latent representations of large transformer-based ASR models focusing on the phonetic context. We specifically investigate the phonotactics of onsets and codas in the English language.

# 6 Limitations

The scope of this work is limited to the encoder-decoder version of the transformer-based ASR models. While both encoder-decoder and encoder-only models might show the same probing accuracy for the same task, the capacity and the location (encoder, cross-attention, or decoder blocks) of the relevant information might vary (Mohebbi et al., 2023). Also, the probes are multi-class classifiers which means that a reduction in the performance of one class affects the output probabilities in favour of the other classes.

## Acknowledgments

This research was conducted with the financial support of Science Foundation Ireland under Grant Agreement No. 13/RC/2106_P2 at the ADAPT SFI Research Centre at University College Dublin. ADAPT, the SFI Research Centre for AI-Driven Digital Content Technology, is funded by Science Foundation Ireland through the SFI Research Centres Programme. For the purpose of Open Access, the authors have applied a CC BY public copyright licence to any Author Accepted Manuscript version arising from this submission.

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
