# OpenReview forum: "Uncovering Syllable Constituents in the Self-Attention-Based Speech Representations of Whisper"
_EMNLP/2024/Workshop/BlackBoxNLP — BlackboxNLP 2024_

### Official Review · Reviewer_miFW · 2024-09-06

**Overall Assessment:** 2
**Confidence:** 4

**Best Paper:**

1

**Best Paper Justification:**

-

**Comments Questions Suggestions And Typos:**

- why did you use TIMIT (it is true TIMIT comes with phonetic alignements but those alignements could be obtained on other transcribed dataset using forced-alignement techniques)

**Paper Summary:**

The paper explores whether the  weights of the Whisper ASR model, along with its latent representations, capture the distinctive features of syllable components (onset, nucleus, and coda) as well as broader phonetic categories (vowels, consonants, and silence). Results confirm that the model does capture these features. More precisely, the probing experiments suggest that the earlier layers of the model encode the necessary information for identifying syllable constituents.

**Summary Of Strengths:**

-not so many work that probes a foundation model (Whiper) related to phonetic dimensions (syllable constituents) even though the probing approach itself is not new

-paper is well written

**Summary Of Weaknesses:**

-choice of TIMIT: experiments are made on TIMIT, i’d be curious to know Whipser ASR performance on this dataset (i'd predict near zero WER) ; so I suspect that TIMIT  might be too easy / too controlled for probing a model like Whipser which might have already solved the transcription task on TIMIT (read/controled speech) => which questions the relevance of results obtained; maybe librispeech could have been used instead or a more challenging corpus


	-the results are overall expected so it is not clear what is the real outcome of this paper (since probling technique itself is not new)

---

### Official Review · Reviewer_SyEQ · 2024-09-09

**Overall Assessment:** 4
**Confidence:** 3

**Best Paper:**

1

**Best Paper Justification:**

N.a.

**Comments Questions Suggestions And Typos:**

I noticed that the figures in the paper are relatively small and hard to read. Given that the page limit is not exploited (7 out of 8 pages are used), it might be nice to give a bit more space to the figures.

See weaknesses for other suggestions for improvements.

**Paper Summary:**

The paper presents a probing study on the Automatic Speech Recognition model Whispher (transformer-based and available trained only on English data as well as in multilingual form). The goal of the probing study is to investigate to what degree and how such a model encodes syllable structure. Syllables can be decomposed into distinct constituents and different languages impose different constraints on how syllables can be constructed. The probing approach relies on attention-head pruning and compares the representation of syllable structure (onset, nucleus, coda) with phonetic categories (consonants, vowels, silence) to investigate their interaction. This is particularly relevant, since knowing that a sound is a vowel should also help in identifying syllable nuclei (always vowels). The results show that both features are mainly encoded in lower layers (specific attention-heads could be identified). Interactions between phonetic categories and syllable structure could not be identified. It seems that the temporal sequence of sounds is more indicative of syllable structure than phonetic category.

**Summary Of Strengths:**

The paper is well-motivated and well-argued.

The experimental setup is logical and defined clearly; the descriptions and explanations are easy to follow.

The topic of the study is highly relevant and helps to gain a better understanding of how transformer models encode linguistic structure.

**Summary Of Weaknesses:**

I did not identify real weaknesses.

I have a minor suggestion for improvement: it would be helpful to be a bit more explicit about the fact that the paper focuses on English syllable structure (it is mentioned, but remains somewhat implicit).

One curiosity question for future research/discussion: I wonder if the other languages in the multi-lingual model are impacted by English syllable structure (since the English training data seem to be overrepresented.

---

### Official Review · Reviewer_1SFh · 2024-09-10

**Overall Assessment:** 4
**Confidence:** 3

**Best Paper:**

1

**Best Paper Justification:**

NA

**Comments Questions Suggestions And Typos:**

1. The last sentence in the abstract is not clear to me: the features mostly depend on the temporal knowledge incorporated in specific SA heads rather than on vowel identification.

**Paper Summary:**

This paper investigates the impact of different self-attention heads in differentiating syllable constituents and phonetic categories. The authors used probing and SA pruning to evaluate Whisper. The authors show that general features of syllable constituents in captured in earlier layers and temporal information is more relevant for syllable identification. The work contributes to a better understanding of transformer ASR models in syllable encoding.

**Summary Of Strengths:**

1. The paper adds to growing literature addressing the interpretability of large ASR models, drawn from methods from text-based transformers.

2. The design is clear and well-structured, and the results are analyzed in sufficient depth.

**Summary Of Weaknesses:**

1. I am very surprised that there is no direct relation between nucleus and vowel recall in layer 0, given that the nature of nucleus and vowel is highly similar. I wonder if the results reflect some other aspects of differences irrelevant to phonetic properties, such as the tasks, the number of classification categories etc.

2. Although the paper presents valuable empirical findings on syllabic and phonetic categorization, the analysis is limited to syllabic constituents and very broad phonetic categories, making it difficult to connect to theoretical insights about phonological structures. It would be interesting to evaluate different levels of information in speech, such as more fine-grained phonemic features (e.g. alveolar) or sentential-level intonation/accent to get a clearer picture on the model's internal representation of speech.

---

### Decision · Program_Chairs · 2024-09-20

**Decision:**

Accept

**Comment:**

Reviewers agree on the contribution of this paper to studying the performance improvements in transformer-based ASR models. The experimental design is clear and sound, and the results are well analyzed and explained. While reviewers expressed concerns about the limited scope of the analysis—focusing primarily on syllabic and phonetic categorization—and certain experimental details, this paper nonetheless adds valuable insights to the workshop.